# Effects of PM2.5 Exposure on the ACE/ACE2 Pathway: Possible Implication in COVID-19 Pandemic

**DOI:** 10.3390/ijerph20054393

**Published:** 2023-03-01

**Authors:** Laura Botto, Elena Lonati, Stefania Russo, Emanuela Cazzaniga, Alessandra Bulbarelli, Paola Palestini

**Affiliations:** 1School of Medicine and Surgery, University of Milano-Bicocca, 20900 Monza, Italy; 2FIMP-Federazione Italiana Medici Pediatri, 00185 Rome, Italy; 3POLARIS Centre, University of Milano-Bicocca, 20126 Milan, Italy

**Keywords:** PM2.5, mice model, RAS system, COVID-19

## Abstract

Particulate matter (PM) is a harmful component of urban air pollution and PM2.5, in particular, can settle in the deep airways. The RAS system plays a crucial role in the pathogenesis of pollution-induced inflammatory diseases: the ACE/AngII/AT1 axis activates a pro-inflammatory pathway counteracted by the ACE2/Ang(1-7)/MAS axis, which in turn triggers an anti-inflammatory and protective pathway. However, ACE2 acts also as a receptor through which SARS-CoV-2 penetrates host cells to replicate. COX-2, HO-1, and iNOS are other crucial proteins involved in ultrafine particles (UFP)-induced inflammation and oxidative stress, but closely related to the course of the COVID-19 disease. BALB/c male mice were subjected to PM2.5 sub-acute exposure to study its effects on ACE2 and ACE, COX-2, HO-1 and iNOS proteins levels, in the main organs concerned with the pathogenesis of COVID-19. The results obtained show that sub-acute exposure to PM2.5 induces organ-specific modifications which might predispose to greater susceptibility to severe symptomatology in the case of SARS-CoV-2 infection. The novelty of this work consists in using a molecular study, carried out in the lung but also in the main organs involved in the disease, to analyze the close relationship between exposure to pollution and the pathogenesis of COVID-19.

## 1. Introduction

Particulate matter (PM), as a major component of air pollutants, contains a complex mixture of smoke, dust, and other solid particles, as well as liquid droplets, present in the air [1].

PM differs in size, shape, and chemical composition. Among the various methods of PM classification, the aerodynamic diameter is certainly the one that best defines its property of being transported in the atmosphere and its ability to be inhaled. Based on this parameter, PM is categorized into three classes: coarse particles or PM10 (ranging from 2.5 to 10 µm); fine particles or PM2.5 (smaller than 2.5 µm); ultrafine particles or PM0.1 (UFP, smaller than 0.1 µm) [2,3].

While larger particles show greater fractional deposition in the extra-thoracic and upper tracheobronchial regions, smaller particles (e.g., PM2.5) are mostly deposited in the deep lung [4]. Direct effects may occur via agents that are able to cross the pulmonary epithelium into the circulation, such as possibly soluble constituents of PM2.5 (e.g., transition metals and polycyclic aromatic hydrocarbons, PAHs) [4,5,6,7,8,9].

This subsequently may contribute to a systemic inflammatory state via increased oxidative stress, potentially leading to increased health risk [9].

It is well known that pollution impairs the first line of upper airway defense [10]; thus, people living in an area with high levels of pollutants are more prone to develop chronic respiratory conditions and susceptible to any infective agent [11].

SARS-CoV-2 (Severe Acute Respiratory Syndrome Corona Virus 2) [12] is the pathogen of COVID-19. This disease was first reported in December 2019 in Wuhan, Hubei Province, China, and then spread worldwide. The course of the disease is usually mild, but in many cases, it may require hospitalization, and degenerate into acute respiratory distress syndrome (ARDS) leading even to death.

A significant overlap was observed between increased mortality and morbidity and pollution levels.

Based on this correlation, many epidemiological studies, summarized in an exhaustive review by Marquès and Domingo [13], have investigated a possible relationship between the high level of SARS-CoV-2 lethality and atmospheric pollution.

Lombardy is one of the Italian regions with the highest level of virus lethality in the world and one of Europe’s most polluted areas [11].

Samples of PM2.5 gravimetrically collected during the winter of 2008 in the urban area of Milan (North Italy) were chemically characterized based on the potential toxicological relevance of its components. Milan winter PM2.5 contains high concentrations of pro-oxidant transition metals and PAHs and is mainly composed of particles ranging in size from 40 nm to 300 nm. Although the chemical composition is similar to that of other European cities, the annual levels of PM2.5 in Milan are higher [6].

PM2.5 induces an inflammation state with consequent production of cytokines that activate the pathways mediated by MAPK or JAK-STAT3, which in turn modulate the expression of matrix metalloproteinases (MMPs).

Phosphorylated STAT3 and phosphorylated ERK act as transcription factors at the nuclear level by increasing inflammation and MMPs expression.

MMPs are zinc-dependent endopeptidases that are capable of degrading the matrix, but also perform other functions, such as activation or inactivation of chemokines/cytokines, and are involved in inflammation [14,15].

The relationships between PM2.5 and inflammation have been mentioned in many pulmonary diseases, such as acute lung injury (ALI), asthma, and chronic obstructive pulmonary disease (COPD) [16,17,18]. Lin and collaborators [19] demonstrated, using a mouse model, that PM2.5-induced ALI is regulated by the Renin-Angiotensin System (RAS) and the Angiotensin-Converting Enzyme II/angiotensin 1-7/Mas receptor (ACE2/Ang(1-7)/Mas) axis has a crucial role in the pathogenesis of lung injury.

RAS is an essential endocrine system, strongly related to the cardiopulmonary system and inflammation which, by activating inflammatory factors in the lung, participates in pulmonary injury [20,21].

ACE and ACE2 are enzymes expressed in various organs, and are two key enzymes of RAS, generating two pathways with opposite effects [22].

In the Angiotensin-Converting Enzyme/AngiotensinII/AngII Type I Receptor (ACE/AngII/AT1R) axis, Ang II produced by ACE1 starting from Ang I interacts with the AT1 receptor, inducing the expression of IL-6, TNF-α, and TGF-β1 [23]. These cytokines activate transduction pathways involving STAT3 and ERK, and lead to increased production of MMPs and pro-inflammatory molecules. Consequently, this pathway is pro-inflammatory.

Instead, in the ACE2/Ang(1-7)/Mas axis, Ang 1-7 produced by ACE2 starting from Ang II interacts with MAS, which represses the STAT3 and ERK transduction pathway reducing the expression of MMPs and pro-inflammatory molecules [24]. Therefore, Ang 1-7 acts by inhibiting inflammatory pathways, JAK-STAT, MAPK, and NF-KB, but also activates anti-inflammatory molecules such as IL-10, and protective pathways such as NRF2, against ROS. Consequently, this pathway has an anti-inflammatory role [25].

Interestingly, the initial cell entry phase of the SARS-CoV-2 requires binding of its homo-trimeric spike glycoprotein to the membrane-bound form of angiotensin-converting enzyme 2 (ACE2) on the target cell [26,27].

Then, the relationship between exposure to PM2.5 and SARS-CoV-2 infection seems to actually converge on the RAS, and in particular on ACE2.

ACE2 acts as a cellular receptor of the virus, and the binding leads to the internalization of the complex in the target cell with consequent down-regulation of ACE2 [28].

Therefore, the imbalance of ACE2/ACE levels in COVID-19 and the dysregulated AngII/AT1R axis may partially be responsible for the cytokine storm and the resulting pulmonary damage [29,30].

The loss of the modulatory effect of Ang(1–7), obtained by its binding to the Mas receptor that attenuates inflammatory response, may be a further contributing factor in the hyper-inflammation status of severe cases of COVID-19 [31].

Our previous studies showed that UFP-induced inflammation and oxidative stress are associated with the alteration of COX-2, HO-1, and iNOS levels [32,33].

Lung and systemic inflammation are responsible for many of the severe cases of COVID-19 [34], which may ultimately cause severe respiratory failure, multi-organ dysfunction, and death [35].

The search for possible therapeutic strategies against SARS-CoV-2 is rapidly proceeding. Several potential target therapies have been proposed, including acetylsalicylic acid for its anti-inflammatory, analgesic, antipyretic, and antithrombotic effects [36].

These effects are obtained because ASA inhibits prostaglandin and thromboxane synthesis by irreversible inactivation of cyclo-oxygenase-1 (COX-1) and cyclo-oxygenase-2 (COX-2). Additional actions have been described to explain the ability of ASA to suppress inflammation, including heme oxygenase (HO) expression induction [37] and iNOS acetylation, resulting in the release of nitric oxide [38].

Based on these assumptions, here we examine, in a mouse model, the effects of PM2.5 exposures on ACE, ACE2, COX-2, HO-1, and iNOS in the main organs involved in COVID-19 pathology (lung, heart, liver, and brain), to test the potential close relationship between PM2.5 exposure and disease severity.

## 2. Materials and Methods

### 2.1. Animals

Male BALB/c mice (7–8 weeks old) were purchased from Harlan and housed in plastic cages under controlled environmental conditions (temperature 19–21 °C, humidity 40–70%, lights on from 7:00 a.m. to 7:00 p.m.) where food and water were administered ad libitum. Animal use and care procedures were approved by the Institutional Animal Care and Use Committee of the University of Milano-Bicocca (protocol number: PP10/2008) and were in compliance with the guidelines set by the Italian Ministry of Health (DL 116/92). Invasive procedures were performed under anesthesia, and an attempt was made to minimize animal suffering.

### 2.2. PM Sources and Characterization

Atmospheric winter PM2.5 was collected in Torre Sarca (Milano) and has already been described [6]. The details of the sampling and chemical analysis performed on PM2.5 were described by Perrone et al. [8,39], while the chemical composition of Milano PM2.5 was summarized in Sancini et al. [40].

The filter extractions were conducted by using an ultrasonic bath (Sonica^®^, SOLTEC, Milan, Italy), specifically developed to maximize the detachment efficiency of the fine PM. Particles were extracted from the filters in ultra-pure water with four cycles of 20 min each, then dried into a desiccator and weighed. PM2.5 aliquots were properly diluted in sterile saline, sonicated, vortexed, and immediately instilled in mice.

### 2.3. Dose

The purpose of this study is to analyze the adverse effects of exposure to PM2.5 on the different organs analyzed. For this reason, we reduced the cumulative PM2.5 dose proposed by Happo et al. [7] to 0.3 mg/mouse to apply the same treatment scheme used by Farina et al. [41] and Sancini et al. [40]. Indeed, this protocol allows for an increase in extrapulmonary adverse effects, the lungs being still affected.

### 2.4. Intratracheal PM2.5 Instillation

Animals were randomly divided into two experimental groups: sham (isotonic solution), and PM2.5-treated mice. Five mice for each experimental group were intratracheally instilled.

Male BALB/c mice were exposed to a mixture of 2.5% isoflurane (Flurane, Merial, Toulouse) anesthetic gas and kept under anesthesia for the whole instillation procedure. Intratracheal instillations with 100 µg of PM2.5 in 100 µL of isotonic saline solution or 100 µL of isotonic saline solution (sham) were administered through a MicroSprayer^®^ Aerosolizer system (MicroSprayer^®^ Aerosolizer- Model IA-1C and FMJ-250 High-Pressure Syringe, Penn Century, Wyndmoor, PA, USA), as previously described [42,43,44].

The intratracheal instillation was performed for a total of three instillations on days 0, 3, 6, and 24 h after the last instillation, mice were euthanized with an anesthetic mixture overdose (Tiletamine/Zolazepam-Xylazine and isoflurane), (Figure 1).

### 2.5. Organ Homogenization

Organs (lung, heart, liver, and brain) of sham and PM2.5-treated mice, after being excised quickly, were washed in ice-cold isotonic saline solution, minced, and suspended in 0.9% NaCl plus protease inhibitors cocktail (Complete, Roche Diagnostics S.p.A Milano, Monza, Italy). The samples were then homogenized for 30 s at 11,000 rpm with Ultra-Turrax T25 basic (IKA WERKE) and sonicated for 30 s. All the above procedures were performed on ice. The samples were stored at −20 °C for subsequent biochemical analyses.

### 2.6. Electrophoresis and Immunoblotting

Lung, heart, liver, and brain homogenates of sham and PM2.5-treated mice were analyzed for protein content by quantification with a micro-bicinchoninic acid (BCA) assay (Sigma-Aldrich Cat# B9643, Cat# C2284 St. Louis, MO, USA); then, 30 µg of total proteins for each sample were subjected to SDS-PAGE (10%) followed by Western blot.

Protein analysis was assessed with specific antibodies: ACE2 (2.5 µg/mL) and ACE (0.05 µg/mL) (R&D Systems, Minneapolis, MN, USA), COX-2 (1:250) (BD Transduction Laboratories, Franklin Lakes, NJ, USA), HO-1 (1:1000) (Cell Signaling Technology, Danvers, MA, USA), iNOS (1:300) (Byorbyt, Cambridge, UK). The secondary antibodies were appropriate horseradish peroxidase (HRP)-conjugated rabbit anti-goat (1:4000) and goat anti-rabbit or anti-mouse (1:5000) (Thermofisher Scientific, Milano, Italy).

Immunoreactive proteins were revealed by enhanced chemiluminescence (ECL SuperSignal detection kit, Thermofisher Scientific, Milano, Italy) and semi-quantitative analysis was estimated by ImageQuant™ 800 (GE Healthcare Life Sciences, Milan, Italy), program 1D gel analysis. No blinding was performed.

Staining of total proteins versus a housekeeping protein represents the actual amount of loading more accurately due to minor procedural and biological variations, as demonstrated by recent studies [45,46]. Accordingly, samples were normalized with respect to the total amount of proteins detected by Ponceau staining, allowing a straightforward correction for lane-to-lane variation [45,47]. Each protein was then expressed as a percentage of the sham, which represents the control.

### 2.7. Statistical Analysis

For each parameter measured in sham and PM2.5-treated mice, the means (±standard error of the mean, S.E.) were calculated.

Statistical differences were tested by one-way ANOVA and t-test and were considered significant at the 95% level (*p*-value < 0.05).

## 3. Results

In this project, we analyzed the effects of PM2.5 sub-acute administrations on mouse lungs, heart, liver, and brain, evaluating their possible implications in COVID pathology.

In 2018, Lin and collaborators hypothesized that acute lung injury (ALI) induced by PM2.5 was regulated by RAS, with a crucial role for the ACE2/Ang(1-7)/MAS axis in the pathogenesis of the damage. In fact, the atmospheric particulate, through the activation of pro-inflammatory pathways, is implicated in different respiratory and cardiovascular diseases, and the RAS system is strongly related to the cardiopulmonary system and inflammation.

To test this hypothesis, they studied the ACE2 expression in the lung tissue of mouse models of ALI induced by PM2.5 and found a significant up-regulation of this protein. In addition, following the ACE2 knockdown, they observed an increase in lung levels of p-STAT3 and p-ERK1/2 as well as a reduction in injury recovery and tissue remodeling. These results confirm that ACE2 is closely involved in the pathogenesis of PM2.5-induced ALI, playing a protective role [19]. The increase of ACE2 in the lung, after PM2.5 exposure, was confirmed by [48].

However, the ACE2 protein, besides counter-regulating the inflammatory effects triggered by PM exposure acting as an organ-protective factor, is also the main receptor of SARS-CoV-2, the virus responsible for the COVID-19 pandemic [49].

The dual function of ACE2, together with the overlapping between the geographic distribution of COVID-19 outbreaks and high local pollution levels, led to the hypothesis of a correlation between the PM concentration, viral infection susceptibility, and severity of symptoms [50,51].

Induction of inflammation and oxidative stress was observed in mice exposed to UFP, resulting in increased COX-2, HO-1, and iNOS, not only in the lung and heart [32] but also in the cerebellum and hippocampus [33].

Interestingly, as recently demonstrated, these proteins have been related to the pathogenesis of COVID-19, once again suggesting a close relationship between air pollution and SARS-CoV-2 infection [52,53,54].

Based on this evidence, we analyzed the ACE/ACE2 protein pathway and COX-2, HO-1, and iNOS protein levels in a mouse model after sub-acute PM2.5 exposure, in order to evaluate a possible molecular correlation between air pollution and susceptibility to SARS-CoV-2 infection.

This study was performed in the lungs and other organs involved in the COVID-19 syndrome, such as the heart, liver, and brain. Although it is known that SARS-CoV-2 infection causes respiratory disease, it also induces adverse effects at the extrapulmonary level.

The effects of PM2.5 exposure vary according to the organs analyzed.

In the lung of PM2.5-treated mice, the levels of ACE2 (+ 40%) and COX-2 (+ 40%) increased compared to the sham (Figure 2).

In the heart tissue, PM2.5 treatment induced a significant decrease in ACE and ACE2 (−34% and −27% respectively) while showing a significant increase in COX-2 level (+21%) and HO-1 (+60%), compared to the sham (Figure 3).

PM2.5-treated mice showed increased levels of ACE (+80%) in the liver, compared to sham (Figure 4), resulting in a significant change in the ACE/ACE2 ratio (+83%) (Table 1).

In the brain, as well as in the liver, PM2.5-treated mice showed increased levels of ACE (+39%), compared to sham (Figure 5), resulting in a significant change in the ACE/ACE2 ratio (+40%) (Table 1).

All the other investigated biomarkers were not affected by PM2.5 repeated instillations, in all the organs considered (Figure 2, Figure 3, Figure 4 and Figure 5).

## 4. Discussion

Several studies have shown that the ACE2/Ang(1-7)/MAS axis is critically involved in lung pathophysiological processes. It can antagonize the negative effects mediated by RAS or the ACE/AngII /AT1 axis, such as lung inflammation, fibrosis, pulmonary arterial hypertension, and apoptosis of alveolar epithelial cells, suggesting an anti-inflammatory and organ-protective role of the ACE2 protein which, however, is also the receptor of SARS-CoV-2 [55].

Therefore, the significant ACE2 increase observed in the lungs after PM2.5 sub-acute exposure might favor SARS-CoV-2 pulmonary entry in case of infection.

Furthermore, many inflammatory and oxidative stress mediators are known to be impaired in COVID-19 and are associated with multi-organ damage and poor disease prognosis [56,57].

Following sub-acute exposure to PM2.5, COX-2 increased, indicating an inflammatory state that a possible infection could exacerbate.

The COX-2 up-regulation is typical of viral infections and COVID-19. In particular, SARS-CoV-2 acute respiratory syndrome induces severe tissue damage by releasing “cellular debris”. Both primary infection and accumulation of cellular debris initiate the stress response of the endoplasmic reticulum and increase the regulation of inflammatory enzymes, including microsomal prostaglandin E synthase-1 (mPGES-1) and prostaglandin-endoperoxide synthase 2 (also known such as COX-2), which subsequently produce eicosanoids: prostaglandins (PG), leukotrienes (LT) and thromboxanes (TX). These pro-inflammatory lipids, named autacoids, trigger the cytokine storm, which mediates the widespread inflammation and organ damage found in patients with severe COVID-19 [58].

Instead, subacute treatment with PM2.5 does not induce changes in HO-1 and iNOS levels in the lungs. These data are in agreement with previous in vivo work [40], in which the huge amount of polycyclic aromatic hydrocarbons (PAHs) characterizing the PM2.5 samples increased lung cytochrome expression, in particular the Cyp1A1 and Cyp1B1, responsible for the metabolism of PAHs. However, PAH metabolism within the lungs did not promote an increase in HO-1 levels.

It is possible to speculate that the treatment with PM2.5 in the lung mainly involves the alveolar-capillary barrier. The increase in vascular permeability following endothelial activation would facilitate the translocation of fine particles from the lungs into the bloodstream.

Concerning the heart, several studies have highlighted the ACE2/Ang(1-7)/MAS axis cardioprotective effect against the damage generated by the ACE/AngII/AT1 axis [59]. Ferreira et al. (2001) observed, for the first time, that the activity of Ang(1-7), produced by ACE2, induced a significant reduction in cardiac arrhythmias related to ischemia/reperfusion (anti-arrhythmogenic effect) beyond a post-ischemic heart function improvement. Subsequent studies have highlighted the ability of Ang(1-7) to attenuate cardiac hypertrophy, suggesting an anti-hypertrophic role [55]. Consequently, the significant decrease in ACE2 observed following PM2.5 sub-acute exposure might be related to ACE reduction. Therefore, the ACE level reduction, inducing a decrease in AngII, would make ACE2 “less necessary” but might lead to increased inflammation and impaired heart function, predisposing to greater severity in cases of SARS-CoV-2 infection [27].

As in the lung, COX-2 level significantly increases in treated mice, compared to sham, following PM2.5 sub-acute exposure. This result suggests a highly compromised inflammatory contest in the heart that could degenerate in case of SARS-CoV-2 infection.

Furthermore, in the heart following sub-acute exposure to PM2.5, HO-1 increased, as noted in our previous work [40], in an attempt probably at a protective response.

Numerous studies have reported the beneficial effects of the ACE2/Ang1-7/MAS axis in counteracting steatosis and non-alcoholic inflammation, liver fibrosis, and insulin sensitivity in the liver. These observations are in agreement with the increase in ACE2 observed in chronic liver lesions in animal and human models. Furthermore, Ang(1-7) is known to suppress the growth of hepatocellular carcinoma and angiogenesis [55].

Lubel et al. [60] demonstrated that patients with liver cirrhosis showed remarkably high concentration levels of both plasma Ang(1-7) and AngII. In cirrhotic rat liver, Ang(1-7) significantly inhibited the vasoconstriction induced by intrahepatic AngII, through the NO signaling pathways dependent on eNOS and guanylate cyclase.

Sub-acute exposure to PM2.5, instead, induces an increase in ACE but not ACE2, showing that exposure to PM2.5 in the liver does not activate the anti-inflammatory ACE2 /Ang(1-7)/MAS axis to counteract the increased ACE. This event causes a significant increase in the ACE/ACE2 ratio and, consequently, in the pro-inflammatory pathways, indicating an inflammatory state that could be exacerbated by possible infection. No significant changes in HO-1 and iNOS were observed in the liver.

Finally, ACE2 is present in the brain, predominantly in neurons [61]. In an interesting review, the physiological aspects of the ACE2/Ang(1-7)/MAS axis in different organs were analysed, and in particular the role of Ang(1-7) in the brain. ACE2 appears to be essential in the central nervous system for cardiovascular regulation. Indeed, transgenic mice overexpressing the synapsin promoter-driven human ACE2 exhibit protective phenotypes for cardiovascular disease. This suggests that the balance between Ang(1-7) and AngII in brain regions, which regulates the autonomic nervous system, is critical [52].

The increase in ACE levels, observed following sub-acute exposure to PM2.5, causes a significant increase in the ACE/ACE2 ratio. Alteration of the balance between Ang(1-7) and AngII indicates a compromised situation in the brain following exposure to PM2.5, which can degenerate in the case of SARS-CoV-2 infection, with serious outcomes also at the heart level.

Furthermore, the slight decrease in HO-1 observed in the brain suggests a lower countering power against the inflammatory cascade in the case of SARS-CoV-2 infection [62], since HO-1 exerts a powerful antioxidant effect degrading heme, a pro-inflammatory mediator. Indeed, a lower expression of HO-1, due to different polymorphisms, has been associated with greater difficulty in counteracting SARS-CoV-2-induced inflammation [63].

## 5. Conclusions

Sub-acute exposure to PM2.5 causes alterations in the ACE/ACE2 system, with possible consequences on COVID-19 pathogenesis.

In an in vivo model of male BALB/c mice, PM2.5 exposure causes variations in the ACE2 and/or ACE levels in all the organs considered.

It is known that ACE2 can counteract the pro-inflammatory pathways activated by ACE, acting as an organ-protective factor, but also acts as a receptor for the entry of SARS-CoV-2 into host cells in case of infection. An alteration of the ACE/ACE2 ratio, when in favor of ACE, suggests a greater probability of manifesting severe symptoms under infection due to the pro-inflammatory pathways’ enhancement. In contrast, a condition favoring ACE2 increase can involve greater susceptibility to SARS-CoV-2 entry in case of contact with the virus.

Therefore, exposure to PM2.5 causes organ-specific changes in the ACE/ACE2 pathway. In all the organs analyzed, HO-1 and iNOS never undergo significant changes, except in the heart, where an increase in HO-1 was observed in agreement with our previous work [40], although we observed a significant COX-2 increase in the lungs and heart, and a considerable increment in the brain.

However, COX-2 plays a central role in viral infections. It is known that SARS-CoV-2 induces the over-expression of COX-2 in human cell cultures and mouse systems [64] and that it could be involved in regulating lung inflammation and disease severity.

In the concept of “risk stratification,” living in a polluted environment can significantly increase the possibility of developing a severe form of COVID-19, especially in individuals with predisposing risk factors (diseases, lifestyle, genetics). This concept could at least partially explain the greater lethality of the virus observed in highly polluted areas, including Lombardy.

The novelty of this work is the use of a molecular approach on an in vivo and non-epidemiological model carried out not only at the pulmonary level but also in the primary organs involved in the disease, in order to analyze the close relationship between pollution exposure and the pathogenesis of COVID-19.

It could be interesting to repeat the analyses following exposure to UFP which, given their aerodynamic diameter of less than 100 nm, have greater penetration and higher translocation rates with possibly worse toxicity profiles.

In our opinion,. experimental studies evaluating the role of air pollution in specific populations are urgently needed for a deeper understanding of the mechanisms leading to a worse prognosis.

## Figures and Tables

**Figure 1 ijerph-20-04393-f001:**
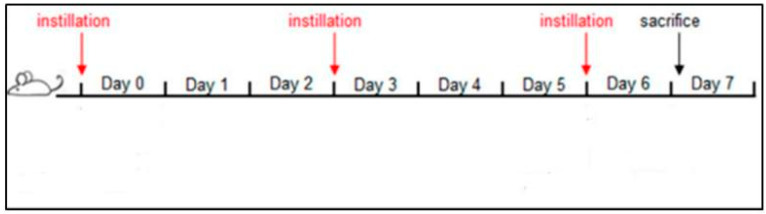
Schematic representation of the subacute treatment of BALB/c mice with PM2.5 [32].

**Figure 2 ijerph-20-04393-f002:**
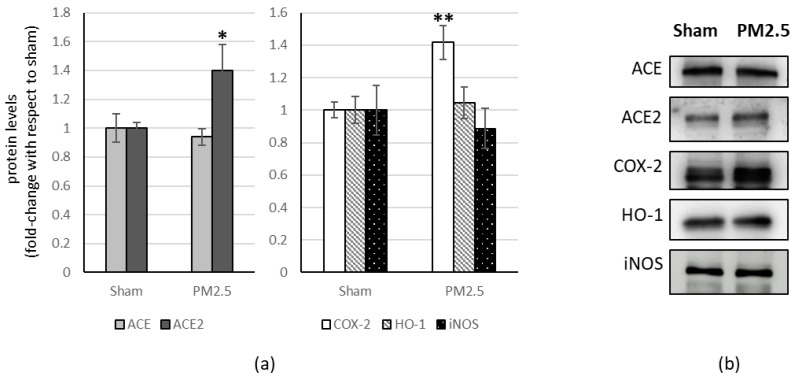
Immunoblotting analysis of ACE, ACE2, and COX-2, HO-1, iNOS in the lung. (**a**) Proteins were evaluated following cell subacute treatment with PM2.5 and have been normalized with respect to the total amount of proteins detected by Ponceau staining, allowing a straightforward correction for lane-to-lane variation. Each protein has been normalized onto the respective sham (n = 5). Values represent Mean ± SE. * *p* < 0.05 versus sham, ** *p* < 0.01 versus sham (**b**) Corresponding representative immuno-blotting images.

**Figure 3 ijerph-20-04393-f003:**
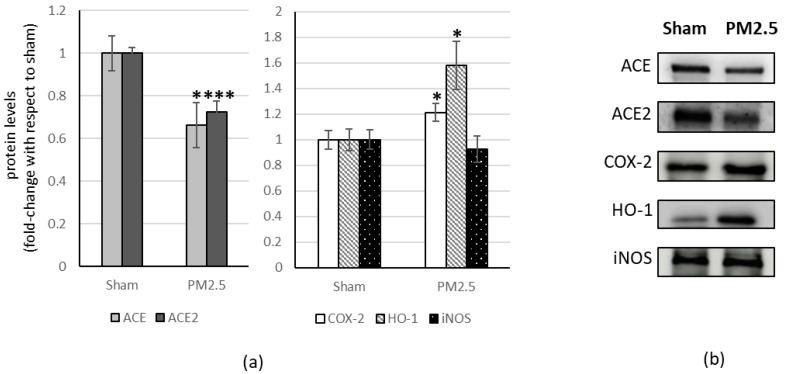
Immunoblotting analysis of ACE, ACE2, and COX-2, HO-1, iNOS in the heart. (**a**) Proteins were evaluated following cell subacute treatment with PM2.5. Each protein has been normalized with respect to the total amount of proteins detected by Ponceau staining, allowing a straightforward correction for lane-to-lane variation. Each protein has been normalized onto the respective sham (n = 5). Values represent Mean ± SE. * *p* < 0.05 versus sham, *** *p* < 0.001 versus sham (**b**) Corresponding representative immuno-blotting images.

**Figure 4 ijerph-20-04393-f004:**
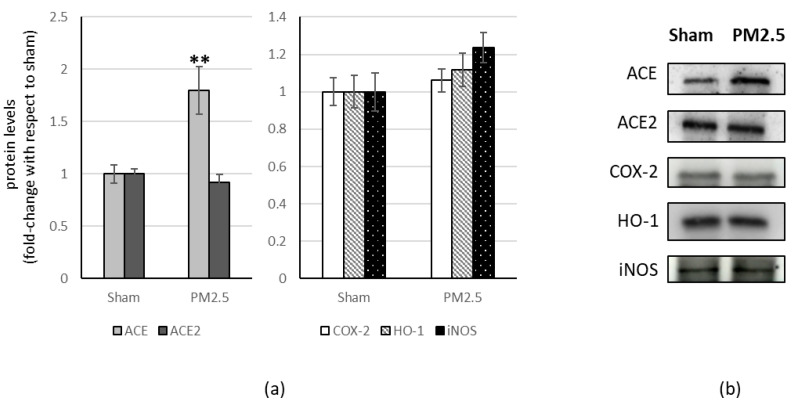
Immunoblotting analysis of ACE, ACE2, and COX-2, HO-1, iNOS in the liver. (**a**) Proteins were evaluated following cell subacute treatment with PM2.5. Each protein has been normalized with respect to the total amount of proteins detected by Ponceau staining, allowing a straightforward correction for lane-to-lane variation. Each protein has been normalized onto the respective sham (n = 5). Values represent Mean ± SE. ** *p* < 0.01 versus sham (**b**) Corresponding representative immuno-blotting images.

**Figure 5 ijerph-20-04393-f005:**
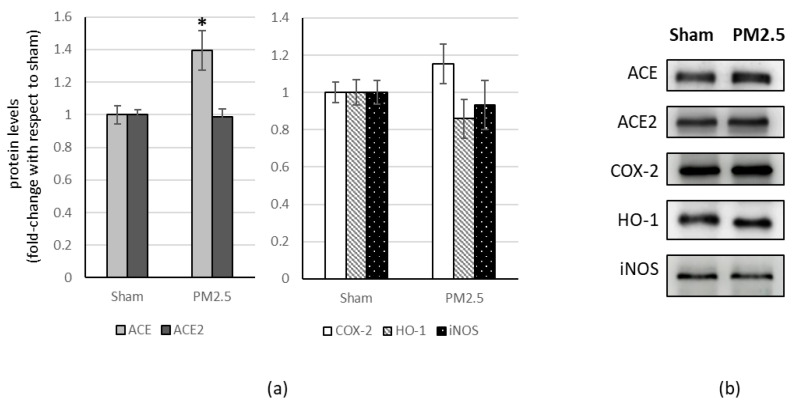
Immunoblotting analysis of ACE, ACE2, and COX-2, HO-1, iNOS in the brain. (**a**) Proteins were evaluated following cell subacute treatment with PM2.5. Each protein has been normalized with respect to the total amount of proteins detected by Ponceau staining, allowing a straightforward correction for lane-to-lane variation. Each protein has been normalized onto the respective sham (n = 5). Values represent Mean ± SE. * *p* < 0.05 versus sham (**b**) Corresponding representative immuno-blotting images.

**Table 1 ijerph-20-04393-t001:** ACE/ACE2 ratio in the various organs analyzed.

Organ	ACE/ACE2	±S.E.	*p*
lung	0.82	±0.087	
heart	0.90	±0.143	
liver	1.83	±0.187	***
brain	1.40	±0.127	*

Each ratio is expressed relative to the sham value. Values represent Mean ± SE. * *p* < 0.05 versus sham, *** *p* < 0.001 versus sham.

## Data Availability

The datasets used and/or analysed during the current study available from the corresponding author on reasonable request.

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
