# Peer review of "Effects of PM2.5 Exposure on the ACE/ACE2 Pathway: Possible Implication in COVID-19 Pandemic"

_ijerph, 2023, doi:10.3390/ijerph20054393_

Round 1
Reviewer 1 Report
The paper is solid overall, but I have a few concerns.
Your finding in the lung (increased ACE2) supports other studies that have shown that both particle air pollution and smoking, increase ACE2.
The measurements of ACE2 in other tissues that are affected by SARS-COV infection, as well as measurements of other inflammatory markers (ACE, COX-2, HO-1 and iNOS) helps complete the picture of how PM2.5 exposure could contribute to more severe COVID-19.
Different tissues had different kinds of changes in inflammatory markers after PM2.5 instillation, so it is difficult to draw firm conclusions about the significance of these findings. The interpretations were generally reasonable, but by necessity a bit speculative. The data do demonstrate PM2.5 instillation does alter the balance of inflammatory mediators, which could be consequential.
I was confused by lines 291 - 294, regarding the lack of an increase in lung HO-1 levels, which was explained by suggesting that the PAHs present in the PM2.5 spread oxidative stress damage "far away from the lungs." Wouldn't you also expect oxidative stress in the lungs, which would activate HO-1?
Author Response
REV1 The paper is solid overall, but I have a few concerns.
Your finding in the lung (increased ACE2) supports other studies that have shown that both particle air pollution and smoking, increase ACE2.
The measurements of ACE2 in other tissues that are affected by SARS-COV infection, as well as measurements of other inflammatory markers (ACE, COX-2, HO-1 and iNOS) helps complete the picture of how PM2.5 exposure could contribute to more severe COVID-19.
Different tissues had different kinds of changes in inflammatory markers after PM2.5 instillation, so it is difficult to draw firm conclusions about the significance of these findings. The interpretations were generally reasonable, but by necessity a bit speculative. The data do demonstrate PM2.5 instillation does alter the balance of inflammatory mediators, which could be consequential.
Thank you for your observations.
Exposure to PM has been related to the clinical severity of people infected by SARS-CoV-2, which can be explained by the oxidative stress and the inflammatory response generated by these particles when entering the respiratory system, as well as by the role of PM in the expression of ACE-2 in respiratory cells in human hosts.
The attachment of SARS-CoV-2 spike glycoprotein with angiotensin-converting enzyme2 (ACE2), as its cellular receptor, triggers complex molecular events that lead to hyperinflammation. At least 10% of the patients with severe COVID-19 will eventually present lung injury, acute respiratory distress syndrome (ARDS), and involvement of multiple organs within 8–14 days of the onset of their illness (Mehdi Mahmudpoura, Jamshid Roozbehb, Mohsen Keshavarza, Shokrollah Farrokhic, Iraj Nabipoura. Cytokine COVID-19 cytokine storm: The anger of inflammation 133 (2020) 155151).
In this context, in our opinion, an approach that takes into account the events that occur at the level of all organs is required to try to understand the interactions and the contribution of each of them to the complex pathogenesis of COVID-19.
Undoubtedly, elucidating the different contributions of this complex network will expand our capacity for treatments to address COVID-19.
I was confused by lines 291 - 294, regarding the lack of an increase in lung HO-1 levels, which was explained by suggesting that the PAHs present in the PM2.5 spread oxidative stress damage "far away from the lungs." Wouldn't you also expect oxidative stress in the lungs, which would activate HO-1?
Regarding lines 291-294, actually, we would have expected an increase in lung HO-1 level following PM2.5 treatment. However, previous research carried out in our laboratory (Sancini et al., 2014) had already shown that the huge amount of PAHs that characterized our PM2.5win samples increased lung cytochrome expression, particularly the Cyp1A1, and Cyp1B1, well-known cytochromes deputized to PAHs metabolism, generating electrophilic metabolites and other reactive oxygen species but PAHs metabolism within the lungs didn’t promote an increase in HO-1 levels. In this situation, we had speculated that the main district involved within the lungs of PM2.5win-treated mice could be the alveolar-capillary barrier. The endothelial activation may therefore promote an increase in vascular permeability, thus facilitating the translocation of fine particles or chemical compounds from the lungs to the bloodstream.
We have added this explanation in the text to improve understanding (lines 324-332).

Reviewer 2 Report
Air pollution is indeed related to the increase in viral and bacterial infections. The authors should include more aspects concerning the relationship between particulate matter and SARS-CoV-2 from a mechanistic point of view. Moreover, could the authors explain that the air pollution levels decreased significantly during the lockdown, but the SARS-CoV-2 transmission was still at high rates?.
The authors found that the expression of ACE2 increased 40% in the mice treated with PM2.5 compared with sham. However, the immunoblotting image of ACE2 is not different from each other, as it does in the image corresponding to COX2, where the expression also increased by 40%. The same happens with the expression of HO-1 in the heart of the mice treated with PM2.5 where the expression increased by 60%; the image does not reflect such an increment.
Author Response
REV2 Air pollution is indeed related to the increase in viral and bacterial infections. The authors should include more aspects concerning the relationship between particulate matter and SARS-CoV-2 from a mechanistic point of view. Moreover, could the authors explain that the air pollution levels decreased significantly during the lockdown, but the SARS-CoV-2 transmission was still at high rates?
Thanks for the question that allows us to clarify the analyzes carried out in our study. As suggested, we added in the introduction more aspects concerning the relationship between particulate matter and SARS-CoV-2 from a mechanistic point of view.
Moreover, in relation to the influence of outdoor air pollutants on the transmission of SARS-CoV-2, the scientific evidence is much limited and the results are conflicting. The hypothesis that the aerosols emitted by an infected person can be deposited on other suspended particles, in particular on the PM, is very controversial and the viability of the virus in PM has not yet been demonstrated (Ana Santurtún, Marina L. Colom, Pablo Fdez-Arroyabe, Álvaro del Real, Ignacio Fernández-Olmo d, María T. Zarrabeitia. Exposure to particulate matter: Direct and indirect role in the COVID-19 pandemic. Environmental Research, https://doi.org/10.1016/j.envres.2021.112261).
Instead, Marquès and Domingo (Marquès and Domingo, 2021) analyzed and reviewed the results of most investigations focused on assessing the influence of various air pollutants on the severity of COVID-19 in patients infected by the coronavirus. They have concluded that there is a significant association between long-term exposure to various outdoor air pollutants: PM2.5, PM10, O3, NO2, SO2, and CO, and the severity/mortality of the disease. Then, prolonged exposure to atmospheric pollution could induce persistent modifications, while short-term changes in air quality may not be sufficient to restore the initial conditions. This might explain the persistent high fatality rate, despite the dramatic reduction of air pollution levels in Lombardy during the lockdown, even considering that there are many contributing factors.
The authors found that the expression of ACE2 increased 40% in the mice treated with PM2.5 compared with sham. However, the immunoblotting image of ACE2 is not different from each other, as it does in the image corresponding to COX2, where the expression also increased by 40%. The same happens with the expression of HO-1 in the heart of the mice treated with PM2.5 where the expression increased by 60%; the image does not reflect such an increment.
The immunoblotting figures are representative of the densitometric data obtained from the analysis of several western blottings.
As suggested, we modified the immunoblotting images so that they were more representative of the results obtained.
